# The Impact of Sphingosine Kinases on Inflammation-Induced Cytokine Release and Vascular Endothelial Barrier Integrity

**DOI:** 10.3390/ijms232112848

**Published:** 2022-10-25

**Authors:** Andreas V. Thuy, Christina-Maria Reimann, Anke C. Ziegler, Markus H. Gräler

**Affiliations:** Department of Anesthesiology and Intensive Care Medicine, Center for Molecular Biomedicine (CMB) and Center for Sepsis Control and Care (CSCC), Jena University Hospital, 07740 Jena, Germany

**Keywords:** sphingosine 1-phosphate, dendritic cell, macrophage, sepsis, systemic infection, interferon-gamma

## Abstract

Sphingosine kinases type 1 and 2 (SphK1/2) are required for the production of the immune modulator sphingosine 1-phosphate (S1P). SphK1 deficient mice (SphK1^−/−^) revealed 50% reduced S1P in plasma, while SphK2^−/−^ mice demonstrated 2–3 times increased S1P levels in plasma. Since plasma S1P is a potent inducer of vascular endothelial cell (VEC) barrier stability, we hypothesized that higher and lower levels of S1P in SphK2^−/−^ and SphK1^−/−^ mice, respectively, compared to wild type (wt) mice should translate into decreased and increased severity of induced systemic inflammation due to improved or damaged VEC barrier maintenance. To our surprise, both SphK1^−/−^ and SphK2^−/−^ mice showed improved survival rate and earlier recovery from inflammation-induced weight loss compared to wt mice. While no difference was observed in VEC barrier stability by monitoring Evans blue leakage into peripheral tissues, SphK1^−/−^ mice demonstrated a distinct delay and SphK2^−/−^ mice an improved resolution of early pro-inflammatory cytokine release in plasma. Ex vivo cell culture experiments demonstrated that bone marrow-derived dendritic cells (BMDC) generated from SphK1^−/−^ and SphK2^−/−^ mice responded with decreased interferon-γ (IFN-γ) production upon stimulation with lipopolysaccharides (LPS) compared to wt BMDC, while activation-induced cytokine expression of lymphocytes and macrophages was not majorly altered. Ex vivo stimulation of macrophages with IFN-γ resulted in increased cytokine release. These results suggest that SphK1/2 are involved in production and secretion of IFN-γ by DC. DC-derived IFN-γ subsequently stimulates the production and secretion of a large panel of inflammatory cytokines by macrophages, which belong to the main cytokine-releasing cells of the early innate immune response. Inhibitors of SphK1/2 may therefore be attractive targets to dampen the early cytokine response of macrophages as part of the innate immune response.

## 1. Introduction

The immune modulator sphingosine 1-phosphate (S1P) is produced by the enzymatic activity of SphK1/2 [1,2]. Single SphK deficient mice are perfectly viable [3,4], while double-knockout of both SphKs in mice is embryonically lethal due to defective blood vessels and severe hemorrhage between embryonic days 11.5 and 13.5 [5]. These results indicate that both SphKs can at least partially compensate each other. On the other hand, single SphK deficient mice can be clearly discriminated by the levels of S1P in blood: while SphK1^−/−^ mice have about 50% reduced S1P levels in blood, blood S1P levels are about 3 times increased in SphK2^−/−^ mice [3,6]. The reduced blood S1P levels in SphK1^−/−^ mice can be explained by reduced S1P production due to the lack of the S1P producing SphK1 [3]. Increased blood S1P levels of SphK2^−/−^ mice were shown to derive from defective SphK2 dependent cellular degradation of blood-borne S1P, particularly in the liver, resulting in S1P accumulation in circulation [6,7].

The different S1P levels in the blood of SphK1^−/−^ and SphK2^−/−^ mice is also indicative of the different cellular functions of both kinases. SphK1 is a cytoplasmic enzyme that is translocated to the cell membrane upon different cell stimuli including immune challenges [8]. This translocation is thought to promote the exportation of S1P into the extracellular space, where it is able to bind and activate S1P receptors in an autocrine or paracrine manner [9]. Thus, SphK1^−/−^ mice demonstrate decreased extracellular levels of S1P due to defective cellular production, particularly of exported S1P. On the other hand, SphK2 is primarily located at the endoplasmic reticulum (ER) [10,11], where many other enzymes of the sphingolipid metabolism are located, including the S1P-lyase [12]. Previous studies demonstrated that deletion of the SphK2 largely prevented the typical cellular increase of S1P after inhibition of the S1P-lyase, which suggested a direct link between the S1P producing enzyme SphK2 and the S1P degrading enzyme S1P-lyase [6]. Subsequent investigations supported this hypothesis by further demonstrating that the SphK2/S1P-lyase pathway is involved in degradation of blood-borne S1P mainly in the liver [7]. Thus, the role of SphK2 is preferentially connected with intracellular metabolism and S1P-lyase-dependent S1P degradation of blood-borne S1P, which also explains the unexpected discrepancy of both single SphK1/2 knockout mice regarding their S1P levels in circulation.

S1P is involved in many different immune modulatory and VEC functions including lymphocyte circulation and VEC barrier stabilization [13,14]. In both cases, blood-borne S1P is an important stimulus for cells that promotes egress of lymphocytes from secondary lymphoid organs into circulation and stabilization of the VEC barrier. Different S1P-levels in blood may, therefore, have an influence on immunity due to modulation of circulating lymphocyte numbers, which should be increased with high S1P levels and decreased with low S1P levels in blood [15]. Concomitantly, VEC barrier integrity may as well be shifted to increased stability with high blood S1P levels and compromised facing low blood S1P levels [16].

In addition to these physiological effects of altered systemic S1P levels, SphK deficiency may also induce local effects in distinct organs and cells. In a model of acute lung injury (ALI), corticosteroids together with an inflammatory stimulus synergistically upregulated SphK1 particularly in macrophages and ameliorated the disease outcome most likely by preventing VEC barrier breakdown [17]. Phosphorylation of sphingosine by SphK1 was also shown to be involved in protease-activated receptor type 1 (PAR1) induced cytokine secretion in dendritic cells (DC) [18]. SphK1 was involved in the progression of D-galactosamine (GalN) and lipopolysaccharide (LPS) induced acute liver failure in mice, where it regulated TNF-α secretion from macrophages [19]. Furthermore, SphK1 was essential for the activation of plasmacytoid DC and production of type I interferon (IFN) as well as pro-inflammatory cytokines stimulated by ligands of toll-like receptor 7 and 9 [20]. On the other hand, SphK2 dampened inflammatory macrophage activation [21]. Thus, SphK1 and SphK2 play multifaceted roles in systemic and local immune responses.

Here, we investigated the role of SphK1 and SphK2 in polymicrobial peritoneal contamination and infection (PCI) as an experimental sepsis model in mice. Surprisingly, both, SphK1^−/−^ and SphK2^−/−^ mice demonstrated alleviated disease progression compared to wild-type (wt) mice despite their opposite modulation of S1P levels in circulation, suggesting that altered systemic S1P levels have little effect on sepsis outcomes. In contrast, deficiency of SphK1 or SphK2 turned out to reduce IFN-γ production and secretion in DC, which further dampened the release of pro-inflammatory cytokines from macrophages. This result confirmed above-mentioned previous reports that revealed a supporting role of both SphKs for cytokine release from DC and macrophages.

## 2. Results

### 2.1. Altered S1P Levels in SphK1/2 Deficient Mice during Sepsis

In order to investigate the role of SphK1 and SphK2 in sepsis, we used the experimental sepsis model of polymicrobial peritoneal contamination and infection (PCI) in mice. For this purpose, wt, SphK1^−/−^ and SphK2^−/−^ mice were systemically infected by intraperitoneal (ip) injection of a defined human stool suspension. Subsequent determination of S1P levels in plasma 6 h, 24 h and 14 d post infection revealed an early drop of 40% (Δ 416 nM) in wt mice, 45% (Δ 163 nM) in SphK1^−/−^ mice and 15% (Δ 362 nM) in SphK2^−/−^ mice (Figure 1A–C). S1P levels started to recover after 24 h and were back to original levels after 14 d. Initial S1P levels in plasma were 3 times lower in SphK1^−/−^ mice and 2.5 times higher in SphK2^−/−^ mice compared to wt mice (Figure 1D). Importantly, these differences in plasma S1P levels between individual mouse strains were preserved after infection (Figure 1E). Tissue levels of S1P were significantly reduced in SphK1/2 deficient mice as depicted here for liver (Figure 1F). There were no significant differences in T cell numbers in blood of wt, SphK1^−/−^ and SphK2^−/−^ mice (Figure 1G).

### 2.2. Improved Survival of SphK1/2 Deficient Mice after Systemic Infection

Previous data suggested that systemic S1P in plasma is important to prevent inflammation-induced vascular leakage, which is also a complication and severity factor in sepsis [13]. If systemic S1P levels were relevant for sepsis severity, SphK1^−/−^ mice and SphK2^−/−^ mice should demonstrate opposite outcomes post infection. Surprisingly, both SphK1^−/−^ mice and SphK2^−/−^ mice revealed improved survival over the time course of 14 d (Figure 2A,B). While only 10% of infected wt mice survived until day 14, the survival rate was 50% of infected SphK1^−/−^ mice and 40% of SphK2^−/−^ mice. The increased survival rate of SphK1/2 deficient mice was accompanied by earlier recovery from sepsis-induced weight loss (Figure 2C). 

### 2.3. Similar Impact of Sepsis-Induced Vascular Leakage in SphK1/2 Deficient Mice

The fact that both SphK1^−/−^ mice and SphK2^−/−^ mice were characterized by increased survival rate upon sepsis induction made us wonder whether the VEC barrier was influenced by different S1P levels in plasma of the respective mice or not. Injection of Evans blue dye in septic mice revealed a significant increase in vascular leakage compared to naïve control mice (Figure 3A). Despite their very low levels of S1P in circulation, naïve SphK1^−/−^ mice did not display any significant basal leakage but exhibited significant barrier disruption post infection similar to wt mice (Figure 3B). Comparable results were obtained with SphK2^−/−^ mice, which also developed significant vascular leakage post infection despite their increased S1P concentration in blood (Figure 3C), demonstrating that S1P levels that were up to three times higher or lower in plasma than normal did not influence vascular barrier integrity under basal and inflamed conditions. Accordingly, the bacterial load in blood and liver was not different between wt, SphK1^−/−^ and SphK2^−/−^ mice 6 h and 24 h post infection (Figure 3D,E).

### 2.4. Delayed Onset and Faster Resolution of Pro-Inflammatory Cytokine Release in SphK1/2 Deficient Mice

To search for potential other explanations for the improved survival rate of SphK1/2 deficient mice, we analyzed the cytokine profile in plasma of wt, SphK1^−/−^ and SphK2^−/−^ mice before and at 6 h, 24 h and 14 d post infection (Figure 4). Levels of the pro-inflammatory cytokines interleukin-6 (IL-6, Figure 4A–C), tumor necrosis factor-alpha (TNF-α, Figure 4D–F) and monocyte chemoattractant protein-1 (MCP-1, Figure 4G–I) peaked 6 h post infection in wt mice, while SphK1^−/−^ mice showed a delayed onset of the pro-inflammatory cytokine response with highest levels 24 h post infection. SphK2^−/−^ mice had overall diminished levels of all investigated pro-inflammatory cytokines that did not significantly alter compared to naïve control levels except for TNF-α at 6 h post infection. The time course of the cytokine release in SphK2^−/−^ mice was similar to wt mice with highest levels 6 h post infection. In contrast to all measured pro-inflammatory cytokines, the anti-inflammatory cytokine IL-10 was highest in all mice 6 h post infection and had a similar profile in wt and SphK1/2 deficient mice (Figure 4J–L). Interferon-γ (IFN-γ) was only detectable at very low levels in plasma 6 h post infection and not different in wt, SphK1^−/−^ and SphK2^−/−^ mice (Figure 4M).

### 2.5. Unaltered Cytokine Release in SphK1/2 Deficient Macrophages, Lymphocytes and Lung Endothelial Cells

To identify the underlying reason for the delayed onset and the reduced amount of cytokines present in plasma of SphK1/2 deficient mice post infection, we investigated the cytokine release of bone marrow-derived macrophages (BMDM), T cells and B cells isolated from spleen, and mouse lung endothelial cells (MLEC) isolated from lung of wt, SphK1^−/−^ and SphK2^−/−^ mice. BMDM, MLEC and B cells were stimulated with 10 μg/mL lipopolysaccharides (LPS), and T cells were activated with 1.6 μg/mL pre-coated α-CD3 and 1 μg/mL soluble α-CD28 antibody 24 h before cytokine measurements. Of the four mainly expressed cytokines, IL-6, IL-10, TNF-α and MCP-1, no significant differences were observed between naïve and LPS-stimulated BMDM from wt, SphK1^−/−^ and SphK2^−/−^ mice except for a slightly increased TNF-α secretion (Figure 5A–D). Secretion of the predominant cytokines IL-6, IL-10, TNF-α and IFN-γ were also not different in naïve and activated T cells from wt, SphK1^−/−^ and SphK2^−/−^ mice (Figure 5E–H). Stimulated B cells from SphK2^−/−^ mice revealed a slightly reduced secretion of IL-6 compared to those isolated from wt and SphK1^−/−^ mice, but at very low total amounts (Figure 5I). Levels of IL-10 and TNF-α were not significantly altered (Figure 5J,K). Similar to the investigated leukocytes, MLEC isolated from wt, SphK1^−/−^ and SphK2^−/−^ mice did not show any significant differences regarding the secretion of IL-6, TNF-α and MCP-1 either (Figure 5L–N).

### 2.6. Reduced IFN-γ Production by LPS-Stimulated DC from SphK1/2 Deficient Mice Dampened Subsequent Cytokine Release by Macrophages

Next, we addressed the cytokine profile of naïve and LPS-stimulated BMDC from wt, SphK1^−/−^ and SphK2^−/−^ mice. While no significant differences were obtained for the secretion of IL-10, TNF-α and IL-1β, minor increases of IL-6 and MCP-1 were observed for LPS-stimulated BMDC from SphK1^−/−^ and SphK2^−/−^ mice compared to wt controls (Figure 6A–E). Importantly, the release of IFN-γ decreased significantly in BMDC derived from SphK1^−/−^ and SphK2^−/−^ mice compared to wt controls (Figure 6F). This result derived from the cytometric bead array (CBA, BD Biosciences, Heidelberg, Germany) 24 h after LPS stimulation and was additionally confirmed by enzyme-linked immunosorbent assay (ELISA) 6 h and 24 h after LPS-stimulation (Figure 6G,H). When added to the mouse macrophage cell lines, RAW264.7 and J774.1, 10 ng/mL IFN-γ significantly increased the release of macrophage-derived MCP-1 (Figure 6I). The release of MCP-1 after addition of 100 ng/mL LPS was slightly higher than the release triggered by 10 ng/mL IFN-γ and did not increase any further after addition of both LPS and IFN-γ (Figure 6I). Thus, reduced release of IFN-γ after activation of DC in SphK1/2 deficient mice likely resulted in hindered subsequent activation of macrophages with consequent delay and/or decrease of a larger panel of macrophage-derived pro-inflammatory cytokines as observed in SphK1^−/−^ and SphK2^−/−^ mice compared to wt mice (Figure 7).

## 3. Discussion

S1P in plasma maintains the VEC barrier under basal and inflamed conditions [13]. Alteration of plasma S1P levels might therefore be an attractive approach to regulate VEC barrier stability. Systemic inflammation often leads to VEC barrier disruption, which is a major contributor to sepsis severity [22]. SphK1^−/−^ mice reveal 2–3 times lower S1P levels in plasma, while SphK2^−/−^ mice demonstrate 2–3 times increased S1P levels in plasma compared to wt mice [3,6]. We therefore tested the influence of altered systemic S1P levels in circulation on VEC barrier stability and sepsis severity by inducing a systemic polymicrobial infection to wt, SphK1^−/−^ and SphK2^−/−^ mice. Clearly, altering S1P levels in plasma up to three times above or below normal levels did not influence barrier maintenance under basal or inflamed conditions.

Nonetheless, SphK1^−/−^ and SphK2^−/−^ mice demonstrated better survival rate than wt mice. Previous reports support the notion that inhibition of both kinases alleviate immune responses by different means. SphK1 expression correlated with the expression of CD14 and mediated LPS-induced hyperinflammation [23]. It was also involved in sepsis-associated liver injury via induction of the intracellular translocation of the high-mobility group box 1 (HMGB1), which, in turn, was associated with a direct interaction of SphK1 and the calcium/calmodulin protein kinase II-δ (CaMKII-δ) [24]. Apigenin, an anti-inflammatory dietary flavonoid, mitigated LPS-induced heart failure by inhibiting SphK1 activity [25]. Intestinal epithelial injury during sepsis was reduced with the non-selective SphK inhibitor *N,N*-dimethylsphingosine (DMS) and SphK1 was shown to be involved in nucleotide-binding oligomerization domain-like receptor containing pyrin domain 3 (NLRP3) inflammasome activation and subsequent lung injury in mice suffering from polymicrobial sepsis [26,27]. The selective SphK2 inhibitor ABC294640 abated graft injury after liver transplantation in rats with concomitantly reduced Toll-like receptor 4 (TLR4) expression, NF-κB activation, pro-inflammatory cytokine secretion, expression of adhesion molecules and infiltration of monocytes/macrophages and neutrophils [28]. On the other hand, SphK2 negatively regulated inflammatory cytokine production in macrophages and resolved lung inflammation by suppressing the stimulator of type 1 interferon gene (STING) in alveolar macrophages via S1P production and secretion in CD11b^+^ macrophages [21,29].

Among the different cellular sources of cytokines that were tested, including lymphocytes, BMDM and MLEC, only SphK1/2 deficient BMDC responded with decreased IFN-γ production upon LPS-stimulation. IFN-γ production by DC after co-stimulation with IL-4/IL-12 and IL-18 was shown before [30,31]. Here, we demonstrate that DC also release IFN-γ in response to LPS treatment and TLR4 activation. DC express TLR4 and are generally capable to respond to LPS [32]. While this is the first report to demonstrate a dampening effect of SphK1/2 deficiency on IFN-γ production in DC, a similar dependency was already shown for other cell types. Treatment of natural killer (NK) cells with the non-selective sphingosine kinase inhibitor SKI resulted in reduced IFN-γ levels following IL-18 stimulation [33]. Inhibition of SphKs also diminished IFN-γ production in T cells [34]. These data support the notion that SphKs are involved in the regulation of IFN-γ production and release in many different immune cells including DC.

Intriguingly, no systematic alterations were observed upon LPS stimulation across different pro-inflammatory cytokines released by SphK1/2 deficient macrophages. Several other studies, however, have previously reported a role of the sphingosine kinases in promoting macrophage inflammatory cytokine releases. These include, among others, studies that have demonstrated reductions in TNF-α release in SphK1^−/−^ mice in azoxymethane-induced models of colon cancer [35], increased M2 macrophage counts and reductions in macrophage release of MCP-1, TNF-α, CXCL-1 and IL-1β in SphK2^−/−^ mice in models of renal injury using primary isolated macrophages [36], reduced TNF-α and IL-6 in models of titanium particle-induced inflammation in SKI-treated RAW264.7 macrophages [37] and reductions in TNF-α concentrations in cultures of SphK1 over-expressing RAW264.7 macrophages infected with *Mycobacterium smegmatis* [38]. Interestingly, all studies with demonstrated reductions in macrophage cytokine responses following SphK depletion or inhibition used models of inflammation that did not rely on the activation of the TLR4 receptor, such as azoxymethane, renal obstruction and titanium particles. Even the *M. smegmatis* infection bypasses TLR4 activation by interacting with TLR2 and MyD88 adaptor proteins. On the other hand, and consistent with our results, LPS treatment of BMDM did not reveal any differences in cytokine production and other inflammatory markers based on the expression of SphKs [39]. Thus, the involvement of specific pathogen recognition receptors (PRR) may be relevant for a direct role of SphKs in macrophage function.

The cytokine profile of wt, SphK1^−/−^ and SphK2^−/−^ BMDM with and without LPS stimulation was not majorly different. With regard to the impact of LPS stimulation on SphK1/2 expression, SphK2 was downregulated in primary human macrophages within 6 h of LPS treatment. Downregulation of SphK2 was crucial for the efficient production of inflammatory cytokines and fostered the induced immune response [21]. Expression of SphK1 was not efficiently induced by LPS but required different synergistically acting co-stimuli for efficient induction like palmitate or glucocorticoids [17,40]. SphK expression also influenced macrophage polarization. Renal inflammation in response to unilateral ureteral obstruction showed reduced renal injury in SphK2^−/−^ mice compared to wt mice, which coincided with a relative increase of anti-inflammatory M2 macrophages [36]. Membrane targeting of SphK1 instead promoted the development of inflammatory M1 macrophages in a mouse model of collagen-induced arthritis. Inhibition of SphK1 or S1P receptors prevented the observed M1 macrophage polarization, indicating a role of SphK1 in the development of M1 macrophages [41].

Macrophages belong to the innate immune cells that are able to trigger a broad cytokine response upon stimulation. Indeed, IFN-γ is such an important stimulus for macrophages that it was originally termed macrophage activating factor (MAF) [42]. In addition, IFN-γ promoted the release of copious amounts of pro-inflammatory cytokines not only from macrophages, but also from lymphocytes and NK cells [43]. In support of our data, pre-incubation of murine macrophage cell lines enhanced the release of cytokines including MCP-1 after LPS stimulation [44]. MCP-1 is a chemoattractant that promoted leukocyte infiltration in inflamed tissues and additionally regulated pro-inflammatory processes mainly by modulating diverse T cell responses [45]. Abrogation of MCP-1 production protected mice against sepsis and endotoxemia [46]. Stimulation of BMDM with mere IFN-γ was already sufficient to induce a robust cytokine response [47]. Interestingly, a previous study investigated cecal ligation and puncture (CLP) as an experimental sepsis model in mice deficient for IFN-γ and found decreased amounts of macrophage inflammatory protein-2 (MIP-2) and IL-6 [48]. These mice were also more resistant to sepsis. A blocking antibody against IFN-γ was partially effective [48]. We therefore suggest that reduced IFN-γ release by SphK1/2 deficient DC dampen the early inflammatory response and cytokine release of macrophages, which, in turn, may have led to the observed increased survival of SphK1^−/−^ and SphK2^−/−^ mice in experimental sepsis. SphK inhibitors may hence be beneficial to dampen the early immune response in systemic infections without compromising the VEC barrier. Further studies are required to decipher the mechanistic involvement of SphKs in IFN-γ release from DC.

## 4. Materials and Methods

### 4.1. Cell Culture

The murine macrophage cell line RAW264.7 (ATCC TIB-71) was cultured in RPMI1640 medium (Thermo Fisher Scientific, Dreieich, Germany) supplemented with 10% fetal bovine serum (FBS, Bio & Sell GmbH, Nürnberg, Germany), 100 U/mL penicillin and 100 μg/mL streptomycin (Thermo Fisher Scientific). The murine macrophage cell line J774.1 (ATCC TIB-67) was cultured in DMEM medium (Thermo Fisher Scientific) supplemented with 10% fetal bovine serum (FBS, Bio & Sell GmbH), 100 U/mL penicillin and 100 μg/mL streptomycin (Thermo Fisher Scientific).

### 4.2. Experimental Sepsis

Polymicrobial peritoneal contamination and infection (PCI) in mice was induced by an intraperitoneal (ip) injection of a human stool suspension. The stool was donated by healthy, non-vegetarian males. Samples were prepared as described [49], aliquoted and frozen at −80 °C. Microbiological analysis revealed a mixture of gram-negative and positive aerobes and anaerobes. Mice were injected with 1 μL/g body weight (BW) intraperitoneally, resulting in a reproducible mortality rate of 90%. Two hours and 14 h post infection, mice received subcutaneous fluid resuscitation with 200 μL 0.9% sodium chloride solution. The BW of the mice was taken before and 6 h, 16 h, 1 d, 2 d, 4 d, 7 d, 9 d and 14 d post infection. SphK1^−/−^ and SphK2^−/−^ mice were kindly provided by Richard Proia (NIH, Bethesda, MD, USA).

### 4.3. Plasma Isolation

Full blood was harvested in heparinized syringes (Braun, Melsungen, Germany) by heart puncture and centrifuged at 4000 relative centrifugal forces (rcf) for 5 min at 4 °C to separate plasma from blood cells.

### 4.4. Isolation of Bone Marrow Cells from Mice

Tibia and fibula were separated and the remaining tissue was mechanically removed. The bones were briefly soaked in 70% ethanol before being submerged in PBS and subsequently in DMEM serum free medium (SFM). The epiphyses were then removed and 15 mL DMEM medium were pushed through the bones via 21-gauge needle attached to a 20 mL syringe (Braun). After the successful harvest of bone marrow cells, single cell suspensions were prepared by pipetting with a 1 mL filter tip pipette (Sarstedt, Nümbrecht, Germany). Cell suspensions were then transferred to a 50 mL Falcon tube (Thermo Fisher Scientific) and centrifuged for 10 min at 400 rcf. Supernatants were aspirated and cells were resuspended in FBS containing 10% DMSO (Carl Roth, Karlsruhe, Germany) and frozen at −80 °C in a Mr. Freeze container (Thermo Fisher Scientific) before transfer to the liquid nitrogen tank.

### 4.5. Isolation of Splenocytes

Spleens were excised from the donor mice and compressed using the blunt end of a 20 mL syringe stopper over a sieve (BD Biosciences). The obtained splenocytes were washed with 10 mL phosphate buffered saline (PBS, Merck, Darmstadt, Germany) and transferred to a 15 mL Falcon tube (Thermo Fisher Scientific). The splenocytes were centrifuged for 5 min at 400 rcf and the supernatant was aspirated. The splenocytes were then resuspended in 10 mL red blood cell (RBC) lysis buffer (4.5 mL 1.54 M NH_4_Cl, 0.5 mL 100 mM KHCO_3_, 10 mM Na2-EDTA, 45 mL H_2_O) to lyse the remaining RBC and centrifuged for 5 min at 400 rcf. The RBC lysis procedure was repeated a second time. The obtained samples were subsequently resuspended in 10% DMSO in FBS and cryopreserved in a Mr. Freeze container before transfer to the liquid nitrogen tank

### 4.6. Separation of B and T Cells

Separation of T and B cells was conducted according to the manufacturer’s protocol (Miltenyi Biotec, Bergisch Gladbach, Germany). Briefly, full cell suspensions of primary murine splenocytes were counted via flow cytometry (FACS) using the Accuri C6 plus (BD Biosciences). First, 5 × 10^7^ cells were removed and centrifuged at 300 rcf for 10 min at 4 °C, the supernatant was then aspirated. The cell pellet was resuspended in 450 μL MACS buffer (2 mM EDTA, 0.5% bovine serum albumin in PBS). For the selection of CD45R (B220) positive cells, 10 μL magnetic beads per 10^7^ cells (50 μL total) was added to each cell suspension, gently vortexed, and incubated for 15 min at 4 °C. During this incubation period, magnetic separation columns were equilibrated by placing columns in the magnetic field and allowing 500 μL MACS buffer to flow through. After the incubation period, cells were washed once by adding 1 mL MACS buffer per 10^7^ cells and centrifuging for 10 min at 300 rcf at 4 °C. The supernatants were then aspirated and the cells were resuspended in 500 μL MACS buffer. The cell suspensions were then added directly to MS separation column, non-magnetically labelled cells were allowed to flow through, and each column was then washed three times with 500 μL MACS buffer. The flow through from each separation was collected. CD45R^+^ cells were harvested by removing the column from the magnetic field, adding 1mL MACS buffer and applying force via the included plunger. The collected cells were then centrifuged for 5 min at 400 rcf, washed once with 10 mL RPMI1640, resuspended in 10 mL RPMI1640 growth medium and counted by FACS. Additionally, the collected flow through was also centrifuged at 300 rcf for 10 min at 4 °C and resuspended in 500 μL MACS buffer for isolation of CD5 (Ly1) positive T-cells. The separation protocol was repeated using the CD5 magnetic beads.

### 4.7. Isolation of Murine Lung Endothelial Cells (MLEC)

C57BL/6J, SphK1^−/−^ and SphK2^−/−^ mice were sacrificed by isoflurane inhalation. Each animal was disinfected by submerging the mouse up to the neck in 70% ethanol. The abdomen was surgically excavated, the lungs were excised and set in ice cold DMEM medium in a 50 mL Falcon tube. The lungs were placed in sterile 90 cm^2^ petri dishes (Sarstedt), and the hila were removed. The lung tissue was minced with surgical scissors (approximately 100 times). The minced tissue was then incubated with 0.1% collagenase in PBS for 1 h at 37 °C in a water bath. Single cell suspensions were prepared by running the minced tissues through a 15 G cannula 20 times and filtered through a steel net. Cells were then centrifuged for 10 min at 200 rcf at room temperature without using a brake. The majority of the media were aspirated, 1 mL was left behind and the cells were resuspended using a 1 mL pipette tip in DMEM/F12 medium (Thermo Fisher Scientific) supplemented with 20% FBS, 1 mM L-glutamine (Thermo Fisher Scientific) and 100 U/mL penicillin. Then, 2 mL additional medium was added and the cells were plated in a 75 cm^2^ flask coated with 0.1% gelatin. The flask was then filled to a final volume of 15 mL medium. Each flask was then placed at 37 °C and 5% CO_2_ and allowed to grow until confluent with media changes every second day. One day after plating, the media were aspirated and a total 15 mL fresh medium was added. Once cells reached confluency, MLEC cells were selected by magnetic bead separation (Dyna Beads coated with anti-CD102, Thermo Fisher Scientific). The magnetic beads were prepared by incubating 2 × 10^6^ beads for 2 h at 4 °C on a rotor with anti-mouse CD102 antibody (7.5 μL per T75 flask). Following incubation, the beads were washed three times with 1 mL 2% FBS in PBS in a magnetic field. Cells were placed on ice and incubated with 100 μL bead suspension per T75 flask for 1 h at 4 °C. Cells were then trypsinized with 2 mL trypsin-EDTA solution (Thermo Fisher Scientific) and washed three times with trypsin. The trypsinization process was inhibited using 6 mL SFM. A 15 mL Falcon tube was placed in the magnetic field, the cells were allowed to pass and were incubated for 10 min. The tube was then removed from the magnetic field and the cells were resuspended in 1ml medium. Cells were then plated onto fresh gelatin-coated flasks and allowed to reach confluency. Once confluent, positive cell selections were repeated as described above. After selection, cells were counted and plated directly onto 12-well cell culture plates (Sarstedt) at a concentration of 1.25 × 10^5^ cells in 1 mL medium.

### 4.8. Preparation of L929 Conditioned Medium

Macrophage-colony stimulating factor (M-CSF) producing L929 cells (ATCC CCL-1) were cultured in DMEM supplemented with 10% FBS, 100 U/mL penicillin, 100 μg/mL streptomycin, 1 mM sodium pyruvate and 1 mM L glutamine (Thermo Fisher Scientific). The L929 cells were cultured until confluency in 5 175 cm^2^ flasks, each with a total volume of 40 mL. The supernatants were collected every second day for a period of 10 d and stored at 4 °C. To remove cell debris, the generated LCCM medium was filtered by passing through a 0.22 μM mesh filter (Carl Roth) under vacuum pressure. Prior to use in macrophage differentiation, 100 mL LCCM medium was diluted in 400 mL DMEM SFM to generate the working solution.

### 4.9. Bone Marrow-Derived Macrophage (BMDM) Differentiation

The cryopreserved bone marrow cells were removed from the liquid nitrogen tank, stored on ice and rapidly thawed at 37 °C in a water bath. One milliliter of each cell suspension was added to 9 mL DMEM SFM. The cells were centrifuged for 10 min at 400 rcf and the supernatant was aspirated. The bone marrow cells were resuspended in 10 mL LCCM medium and transferred to a 175 cm^2^ cell culture flask, 30 mL additional LCCM medium containing M-CSF was then added for a final volume of 40 mL. The cells were allowed to adhere overnight at 37 °C and 5% CO_2_ to remove any unwanted contamination. The suspension cells were isolated by pipetting and the adherent cells were discarded. A total of 10 mL fresh LCCM working medium was added and the harvested cells were plated on either 6-well plates (Sarstedt) for use in the stimulations or in 10 mL on petri dishes for phenotyping. On the fifth day, the media were aspirated and a total of 10 mL fresh differentiation media were added. The differentiation protocol was complete after 7 d and the cells were harvested for either phenotyping or stimulation.

### 4.10. Bone Marrow-Derived Dendritic Cell (BMDC) Differentiation

The cryopreserved bone marrow cells were removed from the liquid nitrogen tank, stored on ice and rapidly thawed at 37 °C in a water bath. One milliliter of each cell suspension was added to 9 mL DMEM SFM. The cells were centrifuged 10 min at 400 rcf and the supernatant was aspirated. The bone marrow cells were resuspended in 15 mL RPMI 1640 medium. The cells were counted by FACS with 10 μL (50 μg/mL stock) propidium iodide (Carl Roth). Then, 3 × 10^6^ bone marrow cells were removed, centrifuged and resuspended in 10 mL RPMI1640 medium containing 20 ng/mL recombinant (rm) IL-4 and 20 ng/mL rm GM-CSF (ImmunoTools, Friesoythe, Germany), 10% FBS, 100 U/mL penicillin and 100 μg/mL streptomycin and plated in a petri dish at 37 °C and 5% CO_2_. On day 3, 10 mL fresh medium was added. On day 6 and 8, 10 mL medium was removed and centrifuged for 5 min at 400 rcf. Throughout the differentiation process, the total volume of each plate was 20 mL. The suspension cells were then resuspended in 10 mL fresh medium and returned to the petri dish. On day 9 the differentiation protocol was complete and the BMDC cells were ready for use in the stimulation experiments or phenotype staining. Only the non-adherent cells were considered BMDCs.

### 4.11. BMDM Stimulation

After the differentiation protocol, the medium was aspirated from each well of the 6-well plates containing the derived BMDM cells. Each well was washed once with 2 mL PBS at 37 °C and 2 mL fresh LCCM medium containing 10 μg/mL lipopolysaccharides (LPS, Merck) was added. The BMDM cells were incubated for 6 h and 24 h periods at 37 °C and 5% CO_2_. At the 6 h time point, 500 μL supernatant was removed and was frozen at −20 °C. Following 24 h incubation, the remaining 1.5 mL was removed and stored at −20 °C.

### 4.12. BMDC Stimulation

Following the differentiation protocol, suspension cells were harvested from the petri dishes by pipetting with a 10 mL pipette. The cells were transferred to a 50 mL Falcon tube. Then, 200 μL cell suspension were removed and used for counting via FACS. BMDC cells were centrifuged at 400 rcf for 5 min. Following the centrifugation process, the supernatants were aspirated and cells were washed once with 10 mL RPMI1640 medium. Cells were resuspended to a concentration of 1 × 10^6^ cells/mL. A total of 1 mL of each cell suspension was added to a well of a 12-well plate (Sarstedt). An additional 1mL medium containing LPS (20 μg/mL = 10 μg/mL final concentration) was added to each well to bring the final volume to 2 mL. The BMDC cells were stimulated for either 6 h or 24 h at 37 °C and 5% CO_2_. Then, 500 μL supernatants were removed following 6 h incubation and the remaining 1.5 mL was removed at the 24 h time point. The supernatants were harvested and centrifuged for 5 min 400 rcf to remove cell pellets. Afterwards, the supernatants were frozen and stored at −20 °C until use.

### 4.13. T Cell Stimulation

The isolated T-cells were stimulated at a concentration of 2 × 10^5^ cells in 200 μL RPMI1640 medium supplemented with 10% FBS, 100 U/mL penicillin and 100 μg/mL streptomycin. Ninety-six-well plates (Sarstedt) were coated prior to stimulation with 75 μL α-CD3 antibody (1.6 μg/mL, ImmunoTools) for 2 h at 37 °C. Each well was washed twice with 200 μL PBS. 2 × 10^5^ cells were added to 100 μL RPMI1640 medium, an additional 100 μL RPMI1640 medium containing α-CD28 antibody (2 μg/mL = 1 μg/mL final concentration, ImmunoTools) was added to respective samples. Only pure medium was added to the control samples. The cells were incubated at 37 °C for 6 h and 24 h. After the incubation period had been completed, the supernatants were removed, and centrifuged for 5 min at 400 rcf to remove cell pellets. The supernatants were stored at −20 °C.

### 4.14. B Cell Stimulation

The isolated B cells were stimulated at a concentration of 2 × 10^5^ cells in 200 μL RPMI1640 medium supplemented with 10% FBS, 100 U/mL penicillin and 100 μg/mL streptomycin. First, 100 μL cell suspension containing 2 × 10^5^ cells was added to each well. Then, 100 μL RPMI1640 medium containing 20 μg/mL LPS (final concentration 10 μg/mL) was added to each well. The control samples remained untreated and received only RPMI medium. The cells were incubated at 37 °C for 6 h and 24 h. After the incubation period had been completed, the supernatants were removed and centrifuged for 5 min at 400 rcf to remove cell pellets. The supernatants were stored at −20 °C. 

### 4.15. MLEC Stimulation

The medium was aspirated from the 12-well plates containing isolated MLEC. Each well was washed once with 2 mL PBS, before 2 mL of fresh medium containing LPS (10 μg/mL) was added to each well. MLEC were incubated for 6 h and 24 h at 37 °C and 5% CO_2_. At the 6 h time point, 500 μL supernatant was removed and frozen at −20°C. At 24 h, the remaining 1.5 mL was removed and stored at −20 °C.

### 4.16. Enzyme-Linked Immunosorbent Assay (ELISA)

Protein measurements were carried out according to the manufacturer’s protocol (Biolegend, San Diego, CA, USA). The capture antibodies were diluted in the supplied coating buffer. First, 100 μL capture antibody was added to each well of a Maxisorp plate (Thermo Fisher Scientific) and was incubated overnight at 4 °C. The plate was then washed four times with 300 μL wash buffer (0.05% Tween-20 in PBS, Carl Roth). To prevent any non-specific binding, each well was blocked with 200 μL of supplied assay diluent and allowed to incubate for at least 1 h on a plate shaker. The plates were then washed four times with 200 μL wash buffer. Standard solutions were prepared by serial dilutions. Then, 100 μL standard or sample was added in duplicate and incubated for 2 h at room temperature on a shaker. The supernatants were aspirated and the plates were washed four times with 200 μL wash buffer. The detection reagents were diluted in assay diluent and 100 μL was added to each well and incubated for 1 h at room temperature on a shaker. The supernatant was removed and the plates were washed four times with wash buffer. Streptavidin-HRP was diluted in assay diluent and 100 μL was added to each well and incubated for 30 min at room temperature on a plate shaker on the medium setting. Each well was washed five times with 200 μL wash buffer for 1 min per wash. The supernatants were removed and 100 μL tetramethylbenzidine (TMB) solution was added to each well and allowed to incubate as described by the manufacturer. The reactions were stopped using 100 μL 1M HCl and 450 nm absorbance was measured using the Multiskan FC plate reader (Thermo Fisher Scientific). All diluents and buffers were prepared according to specific manufacturer’s protocol. 

### 4.17. Cytometric Bead Array (CBA, BD Biosciences)

The CBA used in this study consisted of a panel of seven cytokines/chemokines including: IFN-γ, IL-1ß, IL-6, IL-10, IL-17A, MCP-1 and TNFα. The CBA assay was run in accordance with the manufacturer’s protocol. All mouse plasma samples were pre-diluted 1:4 in assay diluent; cell culture supernatants were diluted as necessary. The capture beads and detection reagents were prepared according to manufacturer’s protocols. Standard curves were prepared via serial 1:2 dilutions from 2500 pg/mL to 9.76 pg/mL; assay diluent alone was used for the blank value. Briefly, 16.7 μL samples/standards and 16.7 μL capture bead solutions were added to 1 mL FACS tubes (Greiner, Frickenhausen, Germany), gently vortexed and allowed to incubate at room temperature for 1 h. After incubation, 16.7 μL detection reagent was added to each sample and allowed to incubate for an additional hour. A total of 300 μL wash buffer was then added and samples were centrifuged in a plate centrifuge for 5 min at 200 rcf. The supernatant was carefully aspirated and the beads were resuspended in 100 μL wash buffer before brief vortexing. The samples were then measured by FACS using the Accuri C6 plus (BD Biosciences) at a flow rate of 35 μL/min and the 2 blue 2 red laser configuration. The filter setting was FL-1 = 533/30, FL-2 = 585/40, FL-3 = 780/60, FL-4 = 675/25. The obtained data were analyzed using the FCAP array software (BD Biosciences). Any samples exceeding the maximum detection range of the assay were further diluted (1:10–1:400 final dilution) and the CBA protocol was repeated. 

### 4.18. Evans Blue Measurement of Vascular Barrier Integrity

Thirty minutes before mice were sacrificed, 100 μL of a 5 mg/mL Evans blue (EB) solution in 0.9% aqueous sodium chloride was injected intravenously (iv). Sacrificed mice were perfused with PBS containing 0.1% heparin (Merck). Afterwards, liver samples were taken, weighted and incubated in 4 times the amount (*v*/*w*) of formamide (Carl Roth) at 60 °C overnight. EB leakage into tissue was determined by analysing the absorption of 100 μL of the formamide supernatant at 620 nm using the Multiskan FC plate reader (Thermo Fisher Scientific).

### 4.19. Lipid Extraction

To extract lipids from 50 μL mouse plasma, 10μL internal standard (30 pmol C17-S1P, Merck) and 950 μL H_2_O were added and transferred into a glass centrifuge tube (VWR, Darmstadt, Germany). After addition of 300 μL 18.5% HCl, 1 mL methanol and 2 mL CHCl_3_, the sample was vortexed for 10 min and subsequently centrifuged for 3 min at 1900 rcf. The lower CHCl_3_ containing phase was transferred to a new glass centrifuge tube. To the remaining aqueous phase, another 2 mL CHCl_3_ was added and vortexing, as well as centrifugation, were repeated. The lower phase was combined with the CHCl_3_ from the first extraction. The CHCl_3_ was vacuum-dried at 60 °C for 45 min in a vacuum rotator (Christ, Osterode, Germany). Subsequently, the lipid sample was resuspended in 100 μL methanol:CHCl_3_ (4:1; *v*:*v*) and vortexed for 1 min. Before analysis, the sample was transferred into HPLC vials (VWR) and stored at −80 °C.

### 4.20. Liquid Chromatography Coupled to Triple-Quadrupole Mass Spectrometry (LC-MS/MS)

For HPLC separation, a 2 × 60 mm MultoHigh reverse phase C18-column with 3 μm particle size (CS-Chromatographie Service GmbH, Langerwehe, Germany) was equilibrated for 5 min with 10% methanol and 90% aqueous solution of 1% formic acid. From 0–0.5 min, the liquid phase was adjusted to 100% methanol. After 1 min, 10 μL of the sample was introduced onto the column. After 15 min, the column was equilibrated with 10% methanol and 90% aqueous solution of 1% formic acid again for the next run. Detection occurred with the QTrap mass spectrometer (Sciex, Framingham, MA, USA) after electrospray ionization (ESI). Transitions were as follows: C17-S1P, *m*/*z* 366/250; S1P, *m*/*z* 380/264. All metabolites were analyzed using the multiple reaction monitoring (MRM) mode. For quantitative analysis, standard curves were generated and analyzed using Analyst 1.6.3 (Sciex).

### 4.21. Flow Cytometry (FACS)

FACS analysis was performed as described [50].

### 4.22. Determination of the Bacterial Load

Measurements of the bacterial load in blood and liver were performed as described [50].

### 4.23. Statistical Analysis

Statistics were calculated using Prism (version 7, GraphPad Software, San Diego, CA, USA). Outliers were identified according to the ROUT method with the desired maximum false discovery rate Q set to 1% [51]. The following *p*-values were used: * *p* < 0.05; ** *p* < 0.01; *** *p* < 0.001.

## Figures and Tables

**Figure 1 ijms-23-12848-f001:**
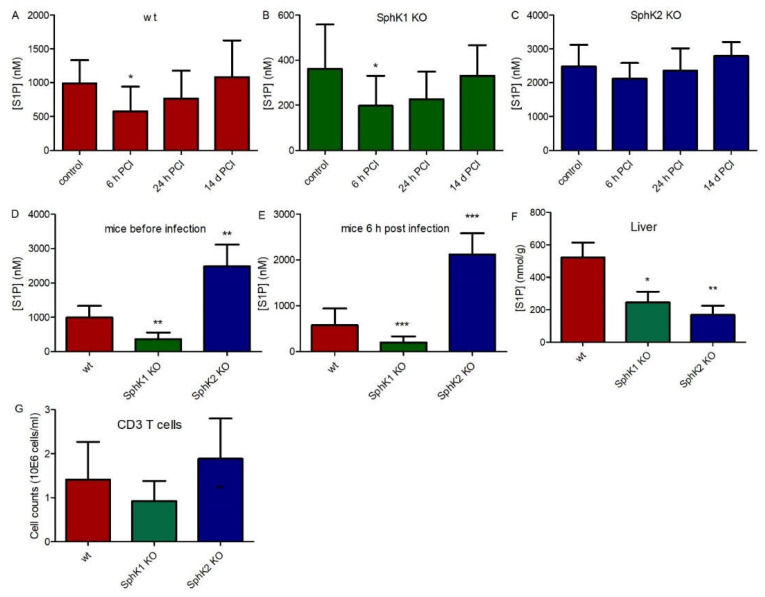
S1P levels in plasma before (control) and 6 h, 24 h and 14 d post infection in (**A**) wt, (**B**) SphK1^−/−^ and (**C**) SphK2^−/−^ mice. Direct comparison of S1P levels in plasma of wt, SphK1^−/−^ and SphK2^−/−^ mice is depicted at the time points (**D**) before infection and (**E**) 6 h post infection in plasma and (**F**) in non-infected liver. (**G**) CD3 T cell counts in wt, SphK1^−/−^ and SphK2^−/−^ mice. Significances were calculated between control and infected mice (**A**–**C**) and wt and SphK1/2 deficient mice (**D**–**G**) using the two-tailed Mann–Whitney U test, *n* ≥ 5, * *p* < 0.05, ** *p* < 0.01, *** *p* < 0.001.

**Figure 2 ijms-23-12848-f002:**
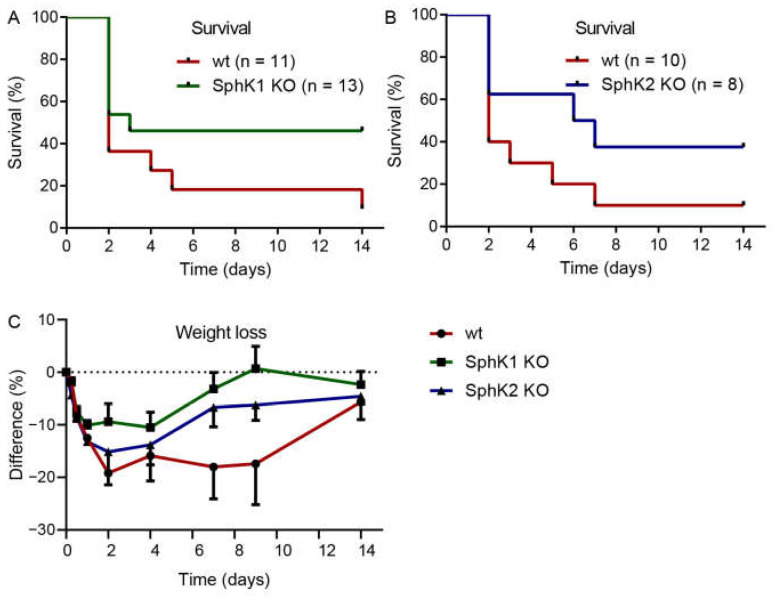
Kaplan–Meier curves demonstrating the survival rate of infected (**A**) wt versus SphK1^−/−^ mice and (**B**) wt versus SphK2^−/−^ mice over the time course of 14 d. Numbers of animals in each group are given in parenthesis. (**C**) Weight loss of wt, SphK1^−/−^ and SphK2^−/−^ mice post infection, *n* = 3–21, depending on the number of surviving mice at each time point.

**Figure 3 ijms-23-12848-f003:**
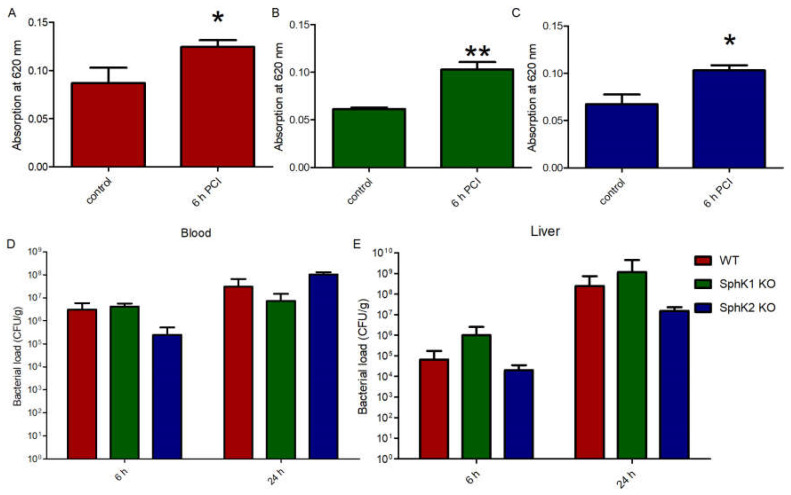
Determination of the VEC barrier stability using Evans blue injection before (control) and 6 h post infection in (**A**) wt, (**B**) SphK1^−/−^ and (**C**) SphK2^−/−^ mice. Bacterial load in (**D**) blood and (**E**) liver 6 h and 24 h post infection in wt, SphK1^−/−^ and SphK2^−/−^ mice. Significances were calculated (**A**–**C**) between control and infected mice or (**D**,**E**) wt and SphK deficient mice using the two-tailed unpaired Student’s *t*-test, *n* ≥ 3, * *p* < 0.05, ** *p* < 0.01.

**Figure 4 ijms-23-12848-f004:**
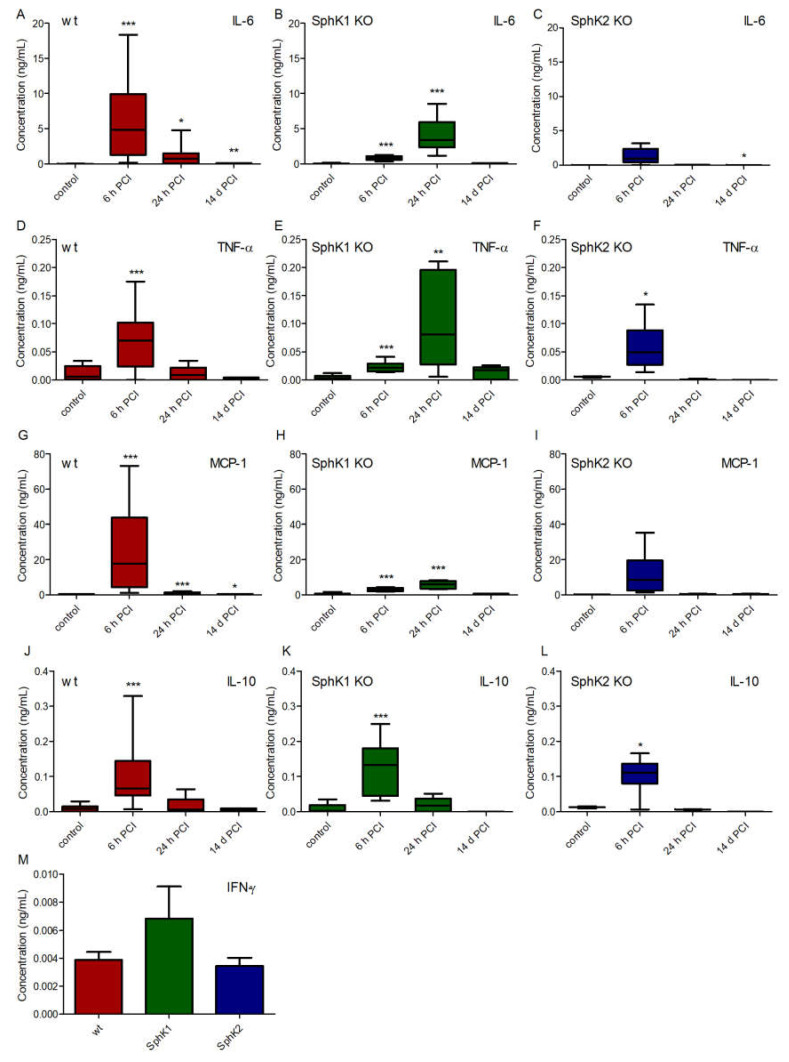
Concentration of (**A**–**C**) IL-6, (**D**–**F**) IL-10, (**G**–**I**) TNF-α and (**J**–**L**) MCP-1 in (**A**,**D**,**G**,**J**) wt, (**B**,**E**,**H**,**K**) SphK1^−/−^ and (**C**,**F**,**I**,**L**) SphK2^−/−^ mice before (control) and 6 h, 24 h and 14 d post infection. (**M**) Concentration of IFN-γ in wt, SphK1^−/−^ and SphK2^−/−^ mice 6 h post infection. Significances were calculated between control and infected mice using the two-tailed unpaired Student’s *t*-test, *n* = 3–21, depending on the number of surviving mice at each time point, * *p* < 0.05, ** *p* < 0.01, *** *p* < 0.001.

**Figure 5 ijms-23-12848-f005:**
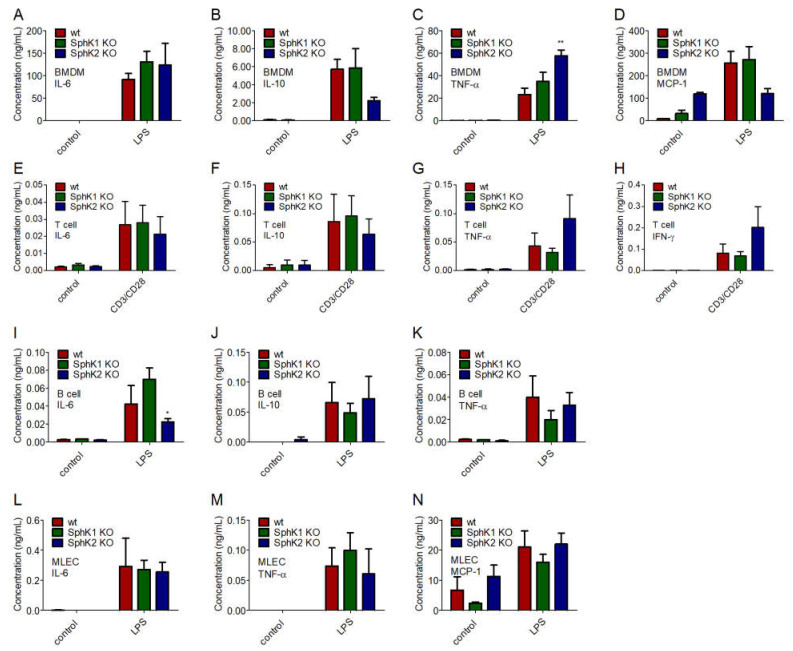
Concentration of (**A**,**E**,**I**,**L**) IL-6, (**B**,**F**,**J**) IL-10, (**C**,**G**,**K**,**M**) TNF-α, (**D**,**N**) MCP-1 and (**H**) IFN-γ in (**A**–**D**) BMDM, (**E**–**H**) T cells, (**I**–**K**) B cells and (**L**–**N**) MLEC unstimulated (control) and 24 h after stimulation with 10 μg/mL LPS or α-CD3 and α-CD28 antibodies. Significances were calculated between cells derived from wt mice and those derived from SphK1/2 deficient mice in the respective control or LPS-treated groups using two-way ANOVA with Bonferroni posttests, *n* ≥ 3, * *p* < 0.05, ** *p* < 0.01.

**Figure 6 ijms-23-12848-f006:**
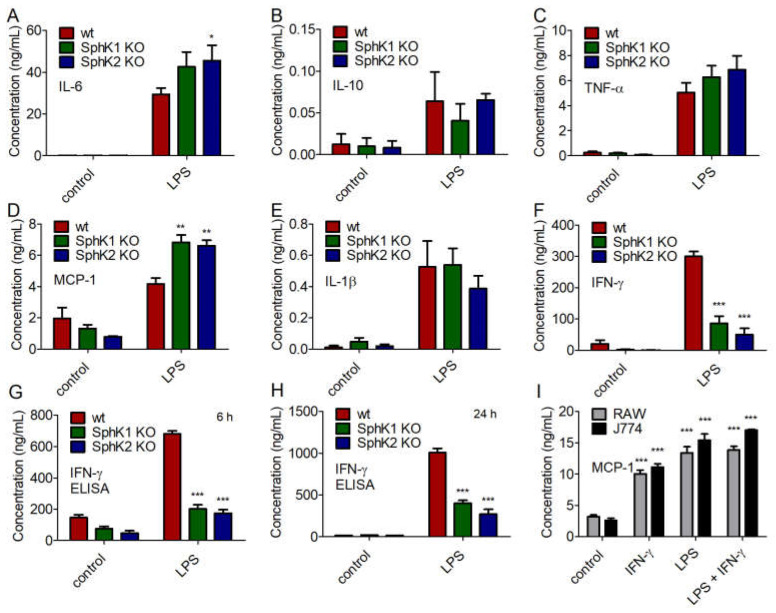
Concentration of (**A**) IL-6, (**B**) IL-10, (**C**) TNF-α, (**D**) MCP-1, (**E**) IL-1β, and (**F**) IFN-γ in BMDC derived from wt, SphK1^−/−^ and SphK2^−/−^ mice unstimulated (control) and 24 h after stimulation with 10 μg/mL LPS, determined by cytometric bead array (CBA, BD Biosciences). Reduced levels of IFN-γ in BMDC derived from SphK1^−/−^ and SphK2^−/−^ mice compared to wt mice as shown in (**F**) were confirmed by ELISA (**G**) 6 h and (**H**) 24 h after stimulation with 10 μg/mL LPS. Significances were calculated between cells derived from wt mice and those derived from SphK1/2 deficient mice in the respective control or LPS-treated groups using two-way ANOVA with Bonferroni posttest, *n* ≥ 3, * *p* < 0.05, ** *p* < 0.01, *** *p* < 0.001. (**I**) Concentration of MCP-1 in RAW264.7 and J774.1 murine macrophage cell line supernatant after overnight stimulation with 10 ng/mL IFN-γ and/or 100 ng/mL LPS. Significances were calculated between control and stimulated cells using the two-tailed unpaired Student’s *t*-test, *n* = 4.

**Figure 7 ijms-23-12848-f007:**
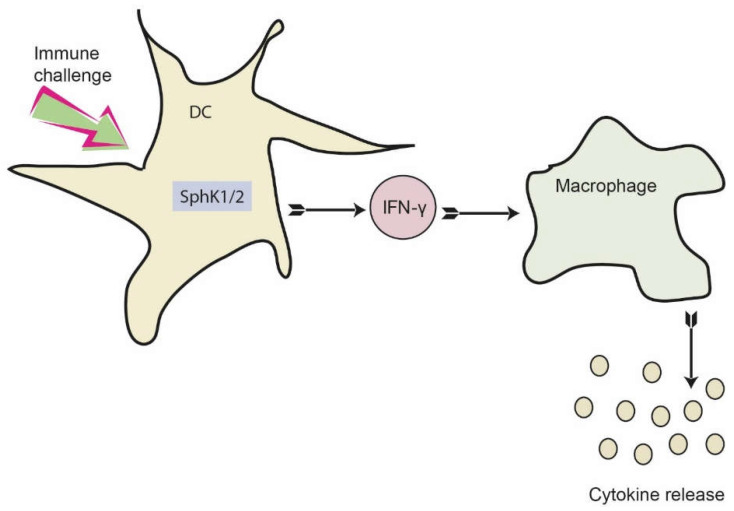
Proposed role of SphKs in DC for dampening the early cytokine response after systemic infection. Activation of SphK1/2 in DC results in increased IFN-γ release, which consequently activates macrophages. As a result, macrophages produce and secrete a number of additional pro-inflammatory cytokines. This eventually results in a dysregulated immune response, which may be prevented by inhibition of SphK1/2 in DC.

## Data Availability

The original contributions presented in the study are included in the article. Further inquiries can be directed to the corresponding author.

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
