# Peer review of "The Impact of Sphingosine Kinases on Inflammation-Induced Cytokine Release and Vascular Endothelial Barrier Integrity"

_ijms, 2022, doi:10.3390/ijms232112848_

Round 1

Reviewer 1 Report

Thuy et al have written a succinct and cogent report describing their interesting finding that both SphK1 and SphK2 contribute to the (counterproductive) burst of inflammatory cytokines and death in response to a bacterial inoculum in mice. They show that, despite differences in the circulating levels of their product S1P in the SphK1 knockout mice (which have low S1P compared to wild type controls) and SphK1 knockout mice (which have over two-fold higher S1P levels than controls), they both exhibit enhanced survival compared to controls in response to sepsis in this model system. They show that vascular permeability is not differentially affected during sepsis. In contrast, upregulation of key cytokines IL-6 and MCP-1 were delayed and had lower amplitude in the SphK1/2 knockout mice, which could not be attributed to differences in the elaboration of cytokines by stimulated SphK1/2 deficient bone marrow derived monocytes, splenic T and B cells and lung endothelial cells. However, they found that bone marrow derived dendritic cells from the knockout mice showed markedly reduced interferon gamma production after stimulation. They further showed that macrophage/monocyte cell lines produced MCP-1 in response to interferon gamma, indicating the latter is an important activator of cytokine responses in mono/macrophage cells. The mechanistic connection hinges on the reduced levels of gamma interferon production by dendritic cells and the inference that this causes a reduction in macrophage derived MCP-1. Overall, the experiments appear robust, the statistical analyses are appropriate, and the writing is excellent.

A minor correction, and a few points the authors might expand on:

1.     In Figure 6 legend, BMDM should be changed to BMDC in the first line.

2.     If IFN gamma-dependent macrophage production of MCP-1 is the main mechanism by which SphK activities contribute to septic shock/death, how do the authors construe the higher levels of MCP-1 in stimulated bone marrow derived dendritic cells from both knockout mice lines?

3.     The role of MCP-1 in general could be spelled out to the reader to emphasize its importance in the immunological response to sepsis.

4.     The authors measured CD3 cells in blood of the three mouse lines under baseline conditions. Did they measure other cell types and in particular the monocytes/macrophages before and after inoculation in the different lines? Did they look at the activation state of the monocytes/macrophages?

5.     The lack of an effect of SphK disruption on barrier integrity in response to bacterial infection is surprising. Might this be due to an accommodation of chronically altered circulating S1P levels, such as an up- or down-regulation of S1P1 on endothelial cells in these mice? If so, an acute inhibition of sphingosine kinase activity in the wild type condition might not be as well tolerated in terms of vascular permeability.

6.     If the vascular permeability and bacterial load are not different in the SphK knockout mice and controls, to what do the authors attribute the difference in survival?

7.     It is interesting that sphingosine phosphate lyase promotes type I interferon production through IKKe (but not through S1P cleavage) and thereby promotes the protective response to viral infection, whereas sphingosine kinases contribute to type II interferon production and thereby promote the counterproductive response to sepsis.

8.     It is also interesting that dendritic cells harbor the lyase activity that controls thymic egress, and dendritic cells also play a key role in the phenomenon described in this study. These findings demonstrate that S1P metabolism is in a pivotal position to influence dendritic cell functions in their roles as sentinels of the immune system.

Author Response

  1. In Figure 6 legend, BMDM should be changed to BMDC in the first line.

Response: Thank you very much for pointing out this mistake, which we corrected in the revised manuscript version.

  1. If IFN gamma-dependent macrophage production of MCP-1 is the main mechanism by which SphK activities contribute to septic shock/death, how do the authors construe the higher levels of MCP-1 in stimulated bone marrow derived dendritic cells from both knockout mice lines?

Response: In this experiment, BMDCs were stimulated with LPS, which is similarly capable to induce MCP-1 production than IFN-gamma, at least in macrophages (Fig. 6I), indicating that LPS-induced MCP-1 production is independent from IFN-gamma stimulation. The increased MCP-1 production by SphK1/2-/- BMDC might be a compensatory mechanism for abrogated IFN-gamma secretion, but this is only speculation and would need further investigation.

  1. The role of MCP-1 in general could be spelled out to the reader to emphasize its importance in the immunological response to sepsis.

Response: Thank you very much for this suggestion. We added the following information to the discussion: “MCP-1 is a chemoattractant that promoted leukocyte infiltration in inflamed tissues and additionally regulated pro-inflammatory processes mainly by modulating diverse T cell responses (46). Abrogation of MCP-1 production protected mice against sepsis and endotoxemia (47).” (p. 19, ll. 527 ff.).

  1. The authors measured CD3 cells in blood of the three mouse lines under baseline conditions. Did they measure other cell types and in particular the monocytes/macrophages before and after inoculation in the different lines? Did they look at the activation state of the monocytes/macrophages?

Response: Unfortunately, we did not measure monocytes/macrophages in our studies because we initially focused on barrier integrity rather than the innate immune response. We tried to determine the activation state of monocytes/macrophages in frozen tissues by qPCR using primers for specific M1 and M2 genes afterwards, but we did not succeed to get a clear picture due to low signal intensities.

  1. The lack of an effect of SphK disruption on barrier integrity in response to bacterial infection is surprising. Might this be due to an accommodation of chronically altered circulating S1P levels, such as an up- or down-regulation of S1P1 on endothelial cells in these mice? If so, an acute inhibition of sphingosine kinase activity in the wild type condition might not be as well tolerated in terms of vascular permeability.

Response: Of course, we cannot exclude the possibility of compensatory mechanisms in the global SphK1/2-/- mice. Unfortunately, we did not check the S1P receptor profile in endothelial cells of the different mouse strains. However, it should be noted that SphK1-/- mice still had about 50 % of plasma S1P levels compared to wt mice. So far, only mice deficient in both SphKs in hematopoietic and vascular endothelial cells demonstrated robust endothelial cell barrier disruption (ref. 13 in manuscript). These mice revealed no detectable S1P in circulation. Thus, it is likely that some amount of S1P is already sufficient to maintain endothelial barrier stability, and that quantitative differences of S1P concentrations in blood have little influence on barrier integrity if basal (low) amounts of S1P are still present.

  1. If the vascular permeability and bacterial load are not different in the SphK knockout mice and controls, to what do the authors attribute the difference in survival?

Response: In fact, the cytokine response of cells might me important for the observed improvement of the survival rate. Lower amounts of pro-inflammatory cytokines IL-6, TNF and MCP-1 at later time points in the disease progression may help to maintain organ function without compromising the immune response against pathogens.

  1. It is interesting that sphingosine phosphate lyase promotes type I interferon production through IKKe (but not through S1P cleavage) and thereby promotes the protective response to viral infection, whereas sphingosine kinases contribute to type II interferon production and thereby promote the counterproductive response to sepsis.

Response: This is, in fact, an interesting result. Noteworthy, the S1P lyase interacted with IKKe in an S1P-independent manner, most likely by direct interaction of the two protein complexes. The impact of SphK1/2 on type II interferon production could be dependent on intracellular S1P production, which was reduced at least in single SphK1 and SphK2 deficient liver tissues. The exact mechanism, however, needs to be further investigated.

  1. It is also interesting that dendritic cells harbor the lyase activity that controls thymic egress, and dendritic cells also play a key role in the phenomenon described in this study. These findings demonstrate that S1P metabolism is in a pivotal position to influence dendritic cell functions in their roles as sentinels of the immune system.

Response: Indeed, it seems that sphingolipid metabolism in dendritic cells might be critically involved in diverse cellular functions. The heterogeneity of different cellular functions might be regulated by distinct intra- and extracellular S1P concentrations as well as direct protein interactions of the enzymes with other complexes like IKKe. Spacial discrimination of S1P concentrations together with non-enzymatic interactions of S1P metabolizing enzymes could be responsible for different cellular responses of the same cells under various conditions.

Reviewer 2 Report

Fig. 7 presents the mecrophage as a black box regarding SphK1/2. This is clearly not the case. LPS alters SphK expression in macrophages, and SphK1/ play a role in macrophage polarization. It is a major weakness of this manuscript that it does not adequately address this point in the discussion.

Author Response

Fig. 7 presents the mecrophage as a black box regarding SphK1/2. This is clearly not the case. LPS alters SphK expression in macrophages, and SphK1/ play a role in macrophage polarization. It is a major weakness of this manuscript that it does not adequately address this point in the discussion.

Response: Thank you very much for pointing out this limitation. We added the following paragraph to the discussion: “The cytokine profile of wt, SphK1-/- and SphK2-/- BMDM with and without LPS stimulation was not majorly different. With regard to the impact of LPS stimulation on SphK1/2 expression, SphK2 was downregulated in primary human macrophages within 6 h of LPS treatment. Downregulation of SphK2 was crucial for the efficient production of inflammatory cytokines and fostered the induced immune response (21). Expression of SphK1 was not efficiently induced by LPS, but required different synergistically acting co-stimuli for efficient induction like palmitate or glucocorticoids (17, 43). SphK expression also influenced macrophage polarization. Renal inflammation in response to unilateral ureteral obstruction showed reduced renal injury in SphK2-/- mice compared to wt mice, which coincided with a relative increase of anti-inflammatory M2 macrophages (39). Membrane targeting of SphK1 was instead promoting the development of inflammatory M1 macrophages in a mouse model of collagen-induced arthritis. Inhibition of SphK1 or S1P receptors prevented the observed M1 macrophage polarization, indicating a role of SphK1 in the development of M1 macrophages (44).” (p. 18, ll. 510 ff.). We did not change Fig. 7, which exclusively shows a brief summary of the results of our own study. We do not want to adorn ourselves with borrowed plumes.